# Transcriptomic and proteomic intra-tumor heterogeneity of colorectal cancer varies depending on tumor location within the colorectum

Sigrid Salling Árnadóttir[1,2], Trine Block Mattesen[1,2], Søren Vang[1,2], Mogens Rørbæk Madsen[3], Anders Husted Madsen[2,3], Nicolai Juul Birkbak[1,2], Jesper Bertram Bramsen[1,2‡], Claus Lindbjerg Andersen[1,2‡]*

1 Department of Molecular Medicine, Aarhus University Hospital, Aarhus, Denmark, 2 Department of Clinical Medicine, Aarhus University, Aarhus, Denmark, 3 Surgical Research Unit, Department of Surgery, Herning Regional Hospital, Herning, Denmark

‡ These authors are joint senior authors on this work.
* cla@clin.au.dk

## Abstract

### Background

Intra-tumor heterogeneity (ITH) of colorectal cancer (CRC) complicates molecular tumor classification, such as transcriptional subtyping. Differences in cellular states, biopsy cell composition, and tumor microenvironment may all lead to ITH. Here we analyze ITH at the transcriptomic and proteomic levels to ascertain whether subtype discordance between multiregional biopsies reflects relevant biological ITH or lack of classifier robustness. Further, we study the impact of tumor location on ITH.

### Methods

Multiregional biopsies from stage II and III CRC tumors were analyzed by RNA sequencing (41 biopsies, 14 tumors) and multiplex immune protein analysis (89 biopsies, 29 tumors). CRC subtyping was performed using consensus molecular subtypes (CMS), CRC intrinsic subtypes (CRIS), and TUMOR types. ITH-scores and network maps were defined to determine the origin of heterogeneity. A validation cohort was used with one biopsy per tumor (162 tumors).

### Results

Overall, inter-tumor transcriptional variation exceeded ITH, and subtyping calls were frequently concordant between multiregional biopsies. Still, some tumors had high transcriptional ITH and were classified discordantly. Subtyping of proximal MSS tumors were discordant for 50% of the tumors, this ITH was related to differences in the microenvironment. Subtyping of distal MSS tumors were less discordant, here the ITH was more cancer-cell related. The subtype discordancy reflected actual molecular ITH within the tumors. The relevance of the subtypes was reflected at protein level where several inflammation markers

Citation: Árnadóttir SS, Mattesen TB, Vang S, Madsen MR, Madsen AH, Birkbak NJ, et al. (2020) Transcriptomic and proteomic intra-tumor heterogeneity of colorectal cancer varies depending on tumor location within the colorectum. PLoS ONE 15(12): e0241148. https://doi.org/10.1371/journal.pone.0241148

**Data Availability Statement:** All relevant data are available on EGA (accession no. EGAS00001004668).

**Funding:** This research was supported by Aarhus University (SSA), The Dagmar Marshalls Foundation (SSA), Aage and Johanne Louis-Hansen's Foundation (SSA), the Novo Nordisk Foundation (NNF16OC0023182) (JBB), the Danish Cancer Society (R40-A1965_11_S2, R56-A3110-12-S2, R107-A7035, R133-A8520), the Danish Council for Independent Research (Medical Sciences) (DFF-0602-02128B, DFF–4183-00619) (CLA), and the National Cancer Institute (R01 CA207467) (CLA). The funders had no role in study design, data collection and analysis, decision to publish, or preparation of the manuscript.

**Competing interests:** The authors have declared that no competing interests exist.

**Abbreviations:** AU, approximately unbiased; CIN, chromosomal instability; CNA, copy number alterations; CMS, consensus molecular subtypes; CRC, colorectal cancer; CRIS, CRC intrinsic subtypes; FPKM, fragments per kilobase million; ITH, intra-tumor heterogeneity; LogCPM, logarithmic counts per million; NPX, normalized protein expression; MSI, microsatellite instability; MSS, microsatellite stability; PEA, proximity extension assay; PIC, protease inhibitor cocktail; PMSF, phenylmethylsulfonyl fluorid; ssGSEA, single sample gene set enrichment analysis; TAM, tumor associated macrophage; Tcyt, Cytotoxic T cell; TME, tumor microenvironment; TT, TUMOR types.

were significantly increased in immune related transcriptional subtypes, which was verified in an independent cohort (Wilcoxon rank sum test; $p<0.05$). Unsupervised hierarchical clustering of the protein data identified large ITH at protein level; as the multiregional biopsies clustered together for only 9 out of 29 tumors.

## Conclusion

Our transcriptomic and proteomic analyses show that the tumor location along the colorectum influence the ITH of CRC, which again influence the concordance of subtyping.

## Introduction

Colorectal cancer (CRC) is one of the leading causes of cancer related deaths worldwide [1]. UICC TNM staging divides the disease in prognostic subgroups, however within each subgroup there is great variability in response to therapy and clinical outcome [2]. In accordance with this, recent molecular studies have shown great inter-tumor heterogeneity within each UICC TNM stage [3–5]. It has been suggested that this heterogeneity complicates development of novel treatment strategies and biomarkers [6, 7].

The heterogeneity may partly be rooted in human embryogenesis, as different embryonic layers give rise to the proximal and the distal colon. Other factors related to the location in the colon may also play a role in forming the heterogeneity, such as the different physiological functions of the proximal colon, the distal colon, and the rectum. Furthermore, there are differences in bacterial composition and basal immune activity from proximal colon to rectum [8–10]. Many CRC features are distinct for tumors of the proximal colon compared to the distal colon or rectum [11] (for reviews see [12–15]). Hypermutated tumors, with microsatellite instability (MSI) are frequently located in the proximal colon, while tumors in the distal colon or rectum commonly exhibit microsatellite stability (MSS), and chromosomal instability (CIN) [16]. At the same time, infiltration of cytotoxic T cells has been linked to a good prognosis in tumors located in the proximal colon, but not for tumors in the distal colon [17].

Besides this inter-tumor heterogeneity, multiregional biopsy studies have shown that CRC commonly exhibit intra-tumor heterogeneity (ITH), which increases the complexity even further. Most studies of ITH have been performed with a cancer-cell-centric focus, with emphasis on genetic sub-clones of cancer cells [18–20]. However, ITH would expectedly also take place on transcriptional level, due to local differences within the tumor both in regard to cancer cells and the tumor microenvironment (TME). While genetic profiles are relatively stable, transcriptional profiles change during cell cycle, cell differentiation, and in response to local signaling. Hence it may be speculated that transcriptional ITH is highly dynamic and that it varies dependent on tumor location.

In the last decade, several methods have been developed for classifying CRC tumors into homogeneous molecular subtypes based on transcriptional profiling. Some of these methods use cancer cell specific transcripts, while others use all transcripts, from immune-, stromal- and cancer cells. Isella and colleagues developed a subtyping approach, the CRC Intrinsic Subtypes (CRIS), which was based solely on cancer cell related transcripts [5]. CRIS subtypes have been reported to be robust across multiregional biopsies, because they are not influenced by the contribution from the TME [21]. The Consensus Molecular Subtypes (CMS) divide CRC tumors into four subtypes without considering the origin of the RNA [3]. Our previously published TUMOR type (TT) classifier also uses all transcripts for classification, but distinguishes

between cancer-, stroma-, and immune cell transcripts [4]. Hence, by combining these three classifiers, they may enhance our understanding of the origin of transcriptional ITH within tumors.

The primary aim of this study was to characterize transcriptional ITH of CRC by sampling multiregional biopsies from each tumor. By combining three subtyping approaches we obtain information about each biopsy from each tumor, and thereby insight into the characteristics of the cancer cells and the TME in each tumor area. This way the subtyping become a useful tool in understanding the origin of ITH and whether it changes depending of the tumor location within the colorectum. The origin of the heterogeneous transcripts was further analyzed through calculations of tumor specific ITH-scores and generation of network maps. In addition we have explored the relation between subtyping, tumor location, and inflammation at protein level.

## Materials and methods

### Study design

The aim of this study was to characterize transcriptomic and proteomic ITH of CRC using multiregional biopsies from primary tumors. From transcriptomic data the degree of ITH, the biology behind ITH, and the impact of tumor location was explored.

### Patient and sample collection

Previously untreated patients diagnosed with TNM stage II and III colorectal tumors larger than 3 cm in diameter were consecutively enrolled at The Surgical Research Unit at Herning Regional Hospital, Denmark in the period from 2014 to 2017. The study was approved by the Central Denmark Regional Committees on Health Ethics (J. no. 1-10-72-221-14), and all patients gave written informed consent. From each tumor, three to five samples were collected from spatially distinct sites of the luminal surface to address ITH. Samples were collected immediately after surgery, snap-frozen in liquid nitrogen, and stored at -80˚ for later analysis. For sample overview see S1 Table.

### MSI status

The MSI status was determined with a pentaplex polymerase chain reaction of quasimono-morphic mononucleotide repeats [22]. Tumors were defined as MSI, if >3 out of 5 PCR markers were positive, as previously described [4].

### RNA purification, sequencing, and data processing

RNA was purified using the RNeasy mini kit (Qiagen, Hilden, Germany). RNA quality was assessed using Agilent RNA 6000 Nano/Pico kits on an Agilent 2100 Bioanalyzer (Agilent Technologies, CA, USA). RNA concentration was measured using Qubit RNA HS assay kit (ThermoFischer Scientific, MA, USA). RNA sequencing was performed at the NGS Core facility, Department of Molecular Medicine, Aarhus University Hospital, as previously described [23]. In short, ribosomal RNA was removed using the Ribo-Zero Gold rRNA Removal Kit (Illumina, CA, USA) leaving both coding and non-coding RNA for whole-transcriptomic sequencing. Synthesis of directional libraries for paired-end sequencing were performed using ScriptSeq v2 RNA-seq Library preparation Kit (Illumina) following manufacturer's instructions. A minimum of 34 million read pairs (median 65 million read pairs) were sequenced per sample on an Illumina NextSeq500 using high output flow cells (Illumina). Data processing of the paired raw sequence reads was performed using TopHat2 [24], with mapping to the

human reference genome HG19. FPKM values were calculated using Cufflinks [25], while raw read counts were calculated using HTSeq and Gencode v19 transcript information [26]. RNA data from four patients, have been published before [18], however, the raw RNA sequencing data have been reanalyzed with an updated pipeline.

RNA sequencing of TNM stage II and III samples from the validation cohort was performed by polyA-sequencing as previously described [4].

## Protein extraction

Proteins were extracted from 10–15 tissue slices of 10μm each. These were transferred to an Eppendorf tube containing 200ul cold RIPA buffer supplemented with 1mM phenylmethylsulfonyl fluorid (PMSF) and 1x protease inhibitor cocktail (PIC)(10x, P8320 Sigma) freshly added (Sigma-Aldrich, MO, USA). After vortexing (1min) the samples were incubated on ice for 30 minutes, followed by centrifugation at 14.000 rpm for 10 minutes at 4˚C. Supernatant was transferred into new cold tubes and stored at -80˚C. Protein quantification was performed using the Pierce™ BCA Protein Assay Kit (ThermoFisher Scientific) following manufacturers protocol. Sample aliquots were diluted in RIPA buffer with PMSF and PIC to obtain 0.4μg/μl. These were transferred into a 96 well plate and stored at -80˚C until analysis.

## Multiplex proximity extension assay (PEA) analysis and quality control

Multiplex PEA analysis was performed by technicians at Olink Proteomics (Uppsala, Sweden). In short, protein samples (0.4μg/μl) in RIPA buffer were mixed with oligonucleotide-labeled antibodies from the Immuno-Oncology panel covering 96 proteins. Each protein is targeted by two antibodies, when these bind their target protein and thereby come into proximity of each other, their oligonucleotides anneal and a PCR target sequence is formed. This was followed by an amplification reaction and subsequently the results were measured using standard qPCR. From the resulting Ct values, normalized protein expression (NPX) values were calculated using the Olink® NPX Manager, which uses both internal and external controls for normalization. All flagged (failed) samples and proteins with an overall detection level below 75% were removed prior to data analysis.

## Clustering and heatmaps

For the RNA heatmap, only protein coding genes were included (gene list obtained online from [27]). Log2(CPM) values were calculated based on TMM normalized raw counts using EdgeR [28], with a filter of log2(CPM)>1. For the protein heatmap, NPX values were used for all proteins from the immuno-oncology panel with confident detection (n = 68). For both approaches, row z-scores were calculated for each gene/protein. Unsupervised clustering and heatmaps were generated using the function *aheatmap* from the Rpackage:NMF [29], with 1-pearson's correlation distance method, and Ward.D2 linkage. Clustering bootstrapping was performed to evaluate the significance of patient-specific clusters using the *pvclust* R package (1000 repetitions) and significance were estimated by Approximately Unbiased (AU) p-values: Clusters were considered significant for AU values ≥95, which indicates a significance p-value ≤0.05 [30].

## Subtype classification

Molecular subtypes were assigned to each sample based on FPKM values. CMS subtypes were assigned using the nearest-centroid single sample predictor CMS classifier and log2 transformed FPKM values [3]. CRIS subtypes were assigned using the CRIS classifier (the

*predictCRIS-ClassKTSP* function) in R (available online from [5]). TT were assigned using the Tumor Subtype Classifier [4]. Caleydo Stratomex was used to visualize the connection between subtypes [31].

### ITH-score and tumor specific network maps

An ITH-score was calculated for each gene based on differences in RNA expression between multiregional biopsies from each tumor. This was done by calculating the standard deviation (STD) of log2(CPM) values for each gene within each tumor. A high ITH-score was defined as STD >0.5. Stroma scores for each gene were defined based on the xenografts studies by Isella *et al.* [32]. Genes with Stroma scores >0.5 were considered "Stromal genes". The gene copy number alterations (CNA) scores were defined for each gene as the standard deviation of the GISTIC2 copy-numbers of all samples within the COREAD cohort available at the UCSC XENA Public Data Hubs [33]. Genes with CNA scores >0.5 were considered "CNA genes" (i.e. affected by chromosomal copy number alternations in CRC). Genes were defined as housekeeping genes if included on the"Human housekeeping genes revisited" list published by Eisenberg et al [34].

Single sample gene set enrichment analysis (ssGSEA) was performed for all samples using the ssGSEAProjection module v 9.1.1 [35] of the Genepattern bioinformatics platform [36] using log2(CPM) values and the Molecular Signatures Database (MsigDB) gene set collection v6.2 as input [35]. For tumors with the largest amount of high ITH-score genes, tumor specific network maps were created using the Enrichment Map app [37] and the Cytoscape software [38]. These network maps were based on inputting the 5000 gene sets with the most varying ssGSEA enrichment scores between multiregional biopsies for each tumor. The generated maps were colored according to differences in ssGSEA enrichment score between biopsies using median normalized values for each gene set.

### Statistical analysis

All statistics were performed in R with functions from the stats package. Tumors were classified as belonging to the left- or right-side cluster in protein heatmap, based on the majority of biopsies. Fisher's exact test was performed using *fisher.test* with default settings. The significance of differences between subtyping groups were tested using the Wilcoxon rank sum test, with the function *wilcox.test*, paired = false.

## Results

### ITH of CRC subtypes varies between proximal and distal tumors

In order to assess how the tumor location affects ITH on the level of transcriptional subtypes, CRC subtyping using the CMS-, CRIS-, and TT-classifiers were applied to total RNA sequencing data from multiregional biopsies (n = 3) from 14 tumors. Since these classifiers use different gene sets and approaches for subtyping, they reflect different kinds of ITH; e.g. ITH arising due to inter-biopsy differences in the cell type distribution of both cancer epithelium and stromal cells (CMS and TT) or ITH arising due to transcriptional differences between the cancer cell populations in the biopsies (CRIS). For MSI tumors from the proximal colon, all three classifiers concordantly called high immune subtypes (CMS1+CRIS-A+SSC) in all biopsies (Fig 1A). For MSS tumors of the proximal colon, the multiregional classifications were less consistent; for these tumors all three classifiers showed inconsistent classification in 50% of tumors, though not necessarily in the same tumors (Fig 1A and 1B). For tumors of the distal colon and rectum, the rate of concordant calls for the multiregional biopsies were much higher

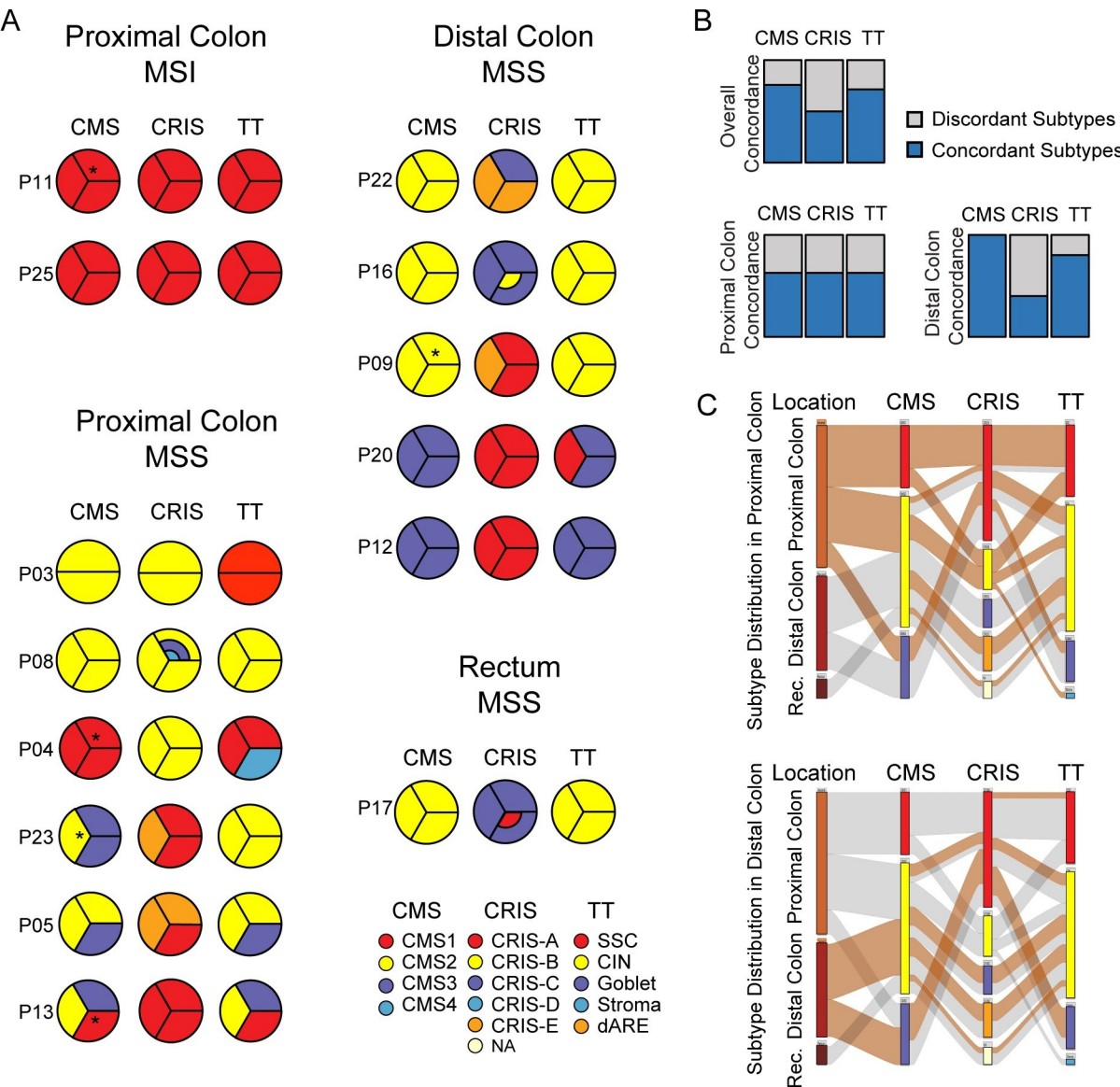

**Fig 1. Tumor location influences ITH of transcriptional CRC subtyping.** (A) All biopsies were classified using three different classifiers; CMS, CRIS and TT. Each circle depicts a tumor, and each piece a biopsy. All tumors are presented three times, one for each classifier. * denotes CMS subtype calls that were only called, when using the SSP.nearest method, in contrast the SSP.predicted method left these samples unclassified. CRIS classifications with multiple colors are due to uncertain subtype calls. (B) Discordant or concordant subtype calls within tumors, for all tumors, proximal tumor, and distal tumors (grey = discordant subtype calls within a tumor, blue = concordant subtype calls). (C) Caleydo Stratomex plots of the distribution and correlation between subtypes called for each biopsy. Top = highlighted for proximal tumors, bottom = highlighted for distal tumors. (CMS = consensus molecular subtypes, CRIS = CRC intrinsic subtypes, TT = tumor types).

(Fig 1A). The CMS classifier showed 100% concordance, the TT classifier 80%, while the CRIS classifier only classified 40% of the tumors concordant (Fig 1A and 1B). The low concordance for the cancer cell transcript based CRIS subtypes indicates that the main reasons for transcriptional ITH in the distal tumors are cancer cell related. The proximal and distal colon tumors differed in their overall subtype distribution and in their composition of subtype calls between classifiers (Fig 1C). The proximal colon tumors were typically more immune-related (CMS1, CRIS-A, CRIS-B, and SSC) than the distal colon tumors (Fig 1C). Furthermore, samples classified as CMS2+CIN, were typically CRIS-B or CRIS-E in the proximal tumors, but CRIS-C or

CRIS-E in the distal tumors. Overall CRIS-E (originally described as Paneth cell-like) was often present in the heterogeneously called tumors (Fig 1A and 1C).

## ITH of overall transcriptional profile

To investigate the overall ITH on gene expression level we performed unsupervised clustering of all tumor biopsies using transcriptomic profiles including coding genes (log2(CPM)>1) (Fig 2A). For 11 out of 14 patients all biopsies clustered together, indicating that the main clustering factor was the tumor of origin (Fig 2A; pair-wise inter-sample correlations in RNA expression profiles are given in S1A Fig). The remaining three tumors that did not cluster in tumor-specific clusters (P05, P13, and P17; indicated by arrows) were among the discordantly subtyped tumors when using the classifiers, particular tumors P05 and P13 (Fig 1A). Hence, the ITH detected with the classifiers, which use a subset of transcripts were also present when assessing ITH across all transcripts. In line with this, tumors that were concordantly subtyped had a significantly higher intra-tumor correlation at transcriptional level, compared to tumors with discordantly subtyped biopsies (p = 0.0037, Wilcoxon rank sum test) (S1B Fig). However, the majority of biopsies cluster in tumor-specific clusters. Furthermore, the biopsies also tend to cluster based on the subtype combination called across the classifiers (Fig 2A). This is especially pronounced for the CMS and TT subtypes while the CRIS subtypes are more intermixed. For some heterogeneous tumors (P05 and P13), their biopsies cluster with biopsies from other tumors with the same subtype combination, rather than clustering in tumor-specific clusters. Looking at all biopsies, they split into three main clusters; the left-most cluster is primarily immune related subtypes (CMS1 + CRIS-A/CRIS-B + SSC). The middle cluster is more mixed; however, all metabolic subtyped (CMS3 + Goblet) biopsies are included within this cluster. The right-most cluster contains subtypes related to chromosomal CNAs (CMS2 + CRIS-C/CRIS-E + CIN) (Fig 2A). Overall, these clusters indicate that both the cancer cell transcripts and the TME contribute to the clustering of the samples.

To explore which types of genes varied the most within cancers, we calculated a tumor specific ITH-score for each gene. This was done using the standard deviation (STD) of the log2(CPM) values for each gene among the multiregional biopsies for the tumor. Furthermore, we did this across biopsies from all tumors to obtain an inter-tumor heterogeneity score. As the TME content may vary between biopsies, we investigated a panel of stromal transcripts to see, whether these showed high ITH-scores. Generally, stromal genes were characterized by having high ITH-scores in most samples (Fig 2B). Furthermore, their expression also showed high inter-tumor variation (Fig 2B: last column). This was different from housekeeping genes, which had lower ITH-scores both within each tumor and between tumors (Fig 2C), which is expected for house-keeping RNA. As CNAs may influence gene transcript levels, we investigated a panel of genes located in genomic regions commonly affected by CNAs in CRC. Expectedly, the inter-tumor variation was larger than the intra-tumor variation (Fig 2D). Generally, the distal tumors had higher levels of ITH in relation to CNA, than the proximal tumors, which may explain the higher number of discordant subtype calls in the distal tumors, when using the cancer cell specific CRIS classifier (Fig 1). However, this was also evident for one proximal tumor (P05), where the CNA related transcripts had a high ITH-score, indicating ITH of CNA events within that tumor (Fig 2D).

## ITH of molecular pathways match heterogeneously called subtypes

The fraction of genes with a high ITH-score (STD > 0.5) varied between tumors (Fig 3A). The tumors whose biopsies did not cluster together (Fig 2A), were among those with the highest number of high ITH-score genes (Fig 3A), and in agreement, the multiregional biopsies of these tumors were often discordantly called by the CRC classifiers (Figs 1A and 3B).

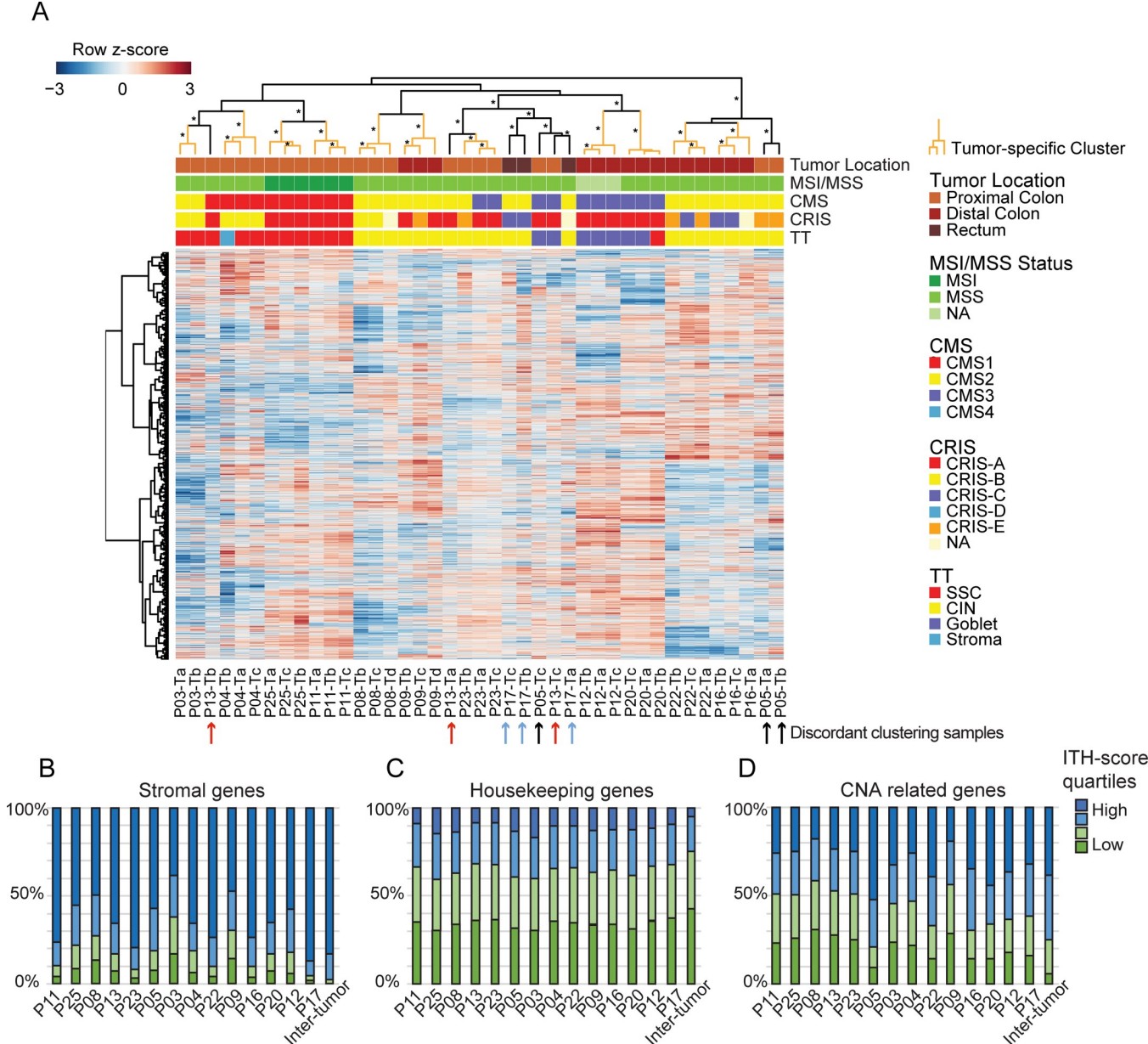

**Fig 2. Transcriptional ITH.** (A) Transcriptional heatmap of protein coding genes. For 11/14 tumors, the multiregional biopsies cluster in tumor-specific clusters (marked with orange). Biopsies from the remaining 3 tumors are combined in clusters (marked in black). The clusters indicated with asterisk (*) were statistical significant as evaluated by bootstrapping (Approximately Unbiased (AU) values ≥95). Arrows below the heatmap indicate discordant clustering biopsies (black: P05, red: P13, blue: P17). Annotations above the heatmap illustrates tumor location, MSI/MSS status, and subtype calls for all three classifiers. The majority of samples cluster according to their subtype combination. (B-D) Distribution of ITH-score for three gene panels, the four colors represent quartiles ranging from low to high ITH-score. Patient specific ITH-scores are calculated as standard deviation for each gene between all biopsies from each tumor. Inter-tumor denotes variation between tumors and is calculated as the STD between all biopsies from all tumors. (B) A panel of stromal genes (n = 618) are enriched for high ITH-score genes. (C) A panel of housekeeping genes (n = 3415) are more stable both within and between tumors. (D) Transcripts related to common copy number alterations (CNAs) are equally distributed in all four categories for most patients (n = 831). P05 have a larger fraction of high ITH-score genes related to CNAs.

Next we investigated if the genes with high ITH-score were involved in specific biological processes. Gene set enrichment analysis, using ssGSEA supported that tumors displaying high ITH on gene level also exhibited high ITH in regards to activity of biological processes. This

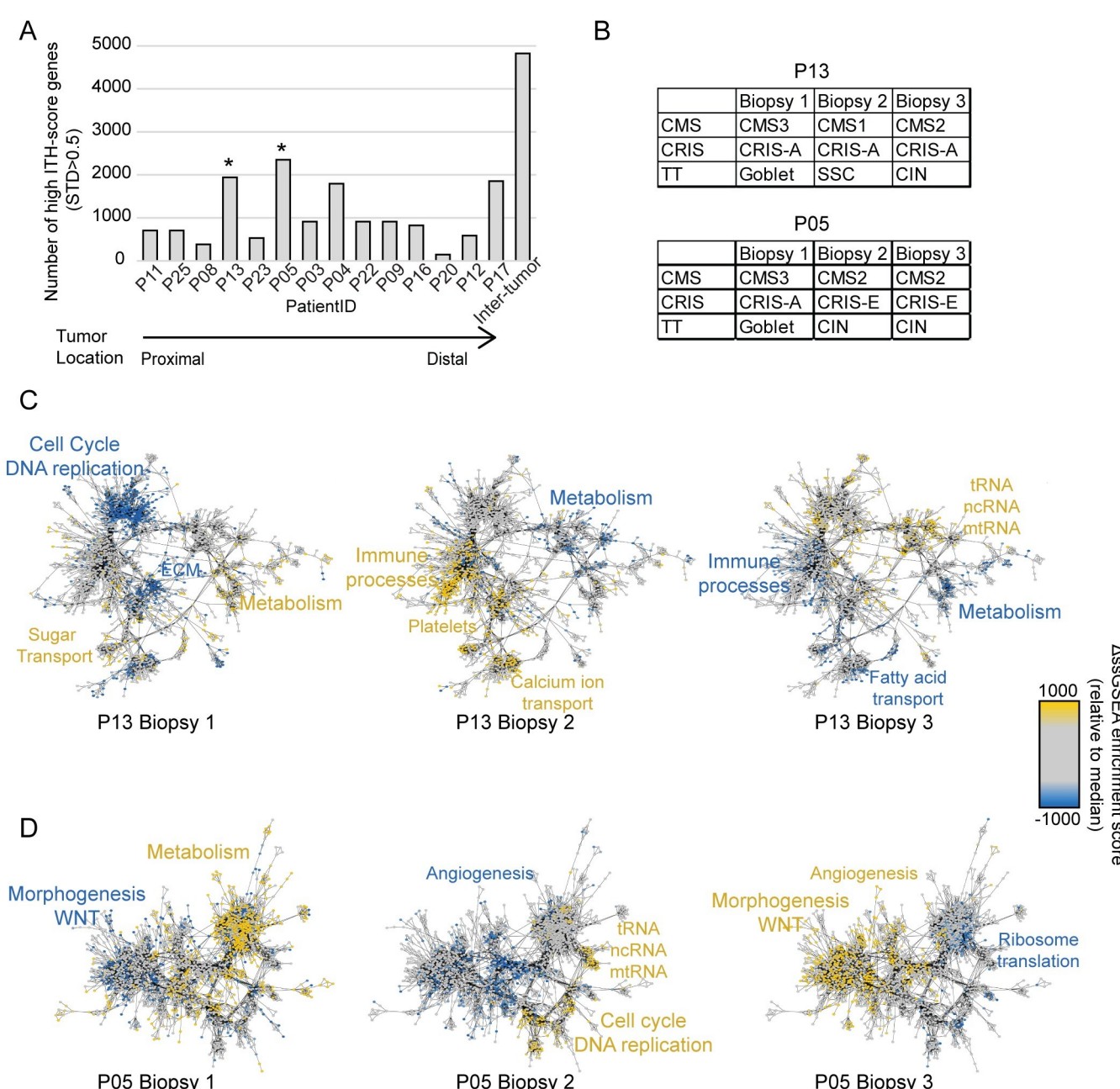

**Fig 3. ITH of molecular pathways.** (A) Barplot with quantification of number of genes with high ITH-score (STD > 0.5) for each tumor and all tumors combined. * denotes tumors with the highest amount of high ITH-score genes, which are included in B-D. (B) Subtyping results for each biopsy from tumor P13 and P05. (C-D) Tumor-specific network maps for two tumors illustrating the 5000 ssGSEA terms with the highest ITH for tumor P13 in (C) and tumor P05 in (D). Yellow dots/font indicates mechanisms that are upregulated in the sample compared to the other samples from the same tumor, while blue indicates downregulated mechanisms.

ITH was plotted in tumor specific network maps, based on up- or down-regulation of the most varying molecular mechanisms (Fig 3C and 3D + S2 Fig). For the two proximal tumors (P05 and P13) with the highest number of high ITH-score genes, the network maps pinpointed biological functions and processes likely underlying the heterogeneous subtype calls (Fig 3C and 3D). Biopsies from the first of these tumors (P13) were classified with three different

classes using both the CMS and TT classifiers (Fig 3B). In the network maps, biopsy 1 had increased activity of metabolic pathways (Fig 3C left), which matched the classified CMS3 and Goblet subtypes. Biopsy 2 was classified as CMS1+ CRIS-A + SSC, all immune related subtypes. This corresponded well with the fact that this biopsy exhibited upregulated immune processes compared to the other biopsies, as well as low metabolic activity (Fig 3C middle). Biopsy 3 was classified as CMS2 + CIN, which matched the low immune and low metabolic activity in this network map (Fig 3C right). Network maps from the second heterogeneous tumor (P05), showed the same tendencies (Fig 3D). Biopsy 1 had higher activity in metabolic gene sets, compared to the other biopsies, which matched the called metabolic subtypes (CMS3 and Goblet; Fig 3D left). Even though biopsy 2 and 3 were classified concordantly, there were some differences in their network maps. Biopsy 2 had increased activity in regards to cell cycle and DNA replication, while biopsy 3 had high activity of Wnt signaling (Fig 3D middle and right). These network maps illustrates the dynamic state of RNA transcripts, as differences in ongoing cellular functions cause ITH on RNA level. Varying biological processes were also observed for the remaining two tumors with high ITH-score genes (P4 and P17) (Fig 2A), even though these were classified concordantly (S2 Fig). These differences included varying activity of processes related to metabolism, extracellular matrix (ECM), cell cycle, and immune processes. Indicating that the transcriptional ITH is only partly captured by the subtyping classifiers.

## Tumor location within the colon influences immune infiltration

Given that stromal RNAs exhibited particularly pronounced ITH (Fig 2B) we explored the ITH of the TME further by measuring the protein levels of 92 cancer and immune related proteins in 29 tumors (3–4 biopsies per tumor) using antibody based PEA analysis (Fig 4). Unsupervised hierarchical clustering of the protein data, identified large ITH at protein level. Only for 9 out of 29 tumors did the multiregional biopsies cluster together. The two main clusters formed were associated with distinct biological and clinical features. The left cluster was significantly enriched for MSI tumors (75%), while the right cluster was enriched for MSS tumors (95%) (p<0.0005, Fisher's exact test) (Fig 4A; pair-wise inter-sample correlations in protein expression profiles are given in S1C Fig).

The proteins analyzed, included several inflammation proteins connected to tumor associated macrophages (TAMs) [39]. The cluster analysis revealed high intra-sample expression correlation between these TAM related inflammation proteins (Fig 4A). The TAM inflammation proteins define a panel, which pinpoints samples on both the left and right sample clusters that are likely to be inflamed (Fig 4A). Likewise, another protein panel with chemokines and granzymes related to an active cytotoxic T (Tcyt) cell response also showed intra-sample expression correlation (Fig 4A).

To analyze the ITH of the TAM inflammation- and Tcyt response-panels within each tumor, we calculated sample means of protein expression for both panels. Some tumors had highly varying levels of each panel between the multiregional biopsies, indicating ITH in relation to the immune activity (Fig 4B). However, it was not the same tumors that exhibited ITH for the two panels. Taking the tumor location into consideration, we observed different patterns for the TAM inflammation panel and the Tcyt cell response panel (Fig 4B). The TAM inflammation levels were highest in MSI tumors, independent of the location within the proximal colon. For MSS tumors, the inflammation levels showed less ITH and overall higher levels in the proximal colon tumors, compared to the distal colon. In contrast, the Tcyt levels were highest in the tumors located in the cecum independent of MSI/MSS status (Fig 4B). This indicates that a certain environment may be present in the cecum, leading to a higher Tcyt cell response in the tumors independent of the mutational profile of the cancer cells [40].

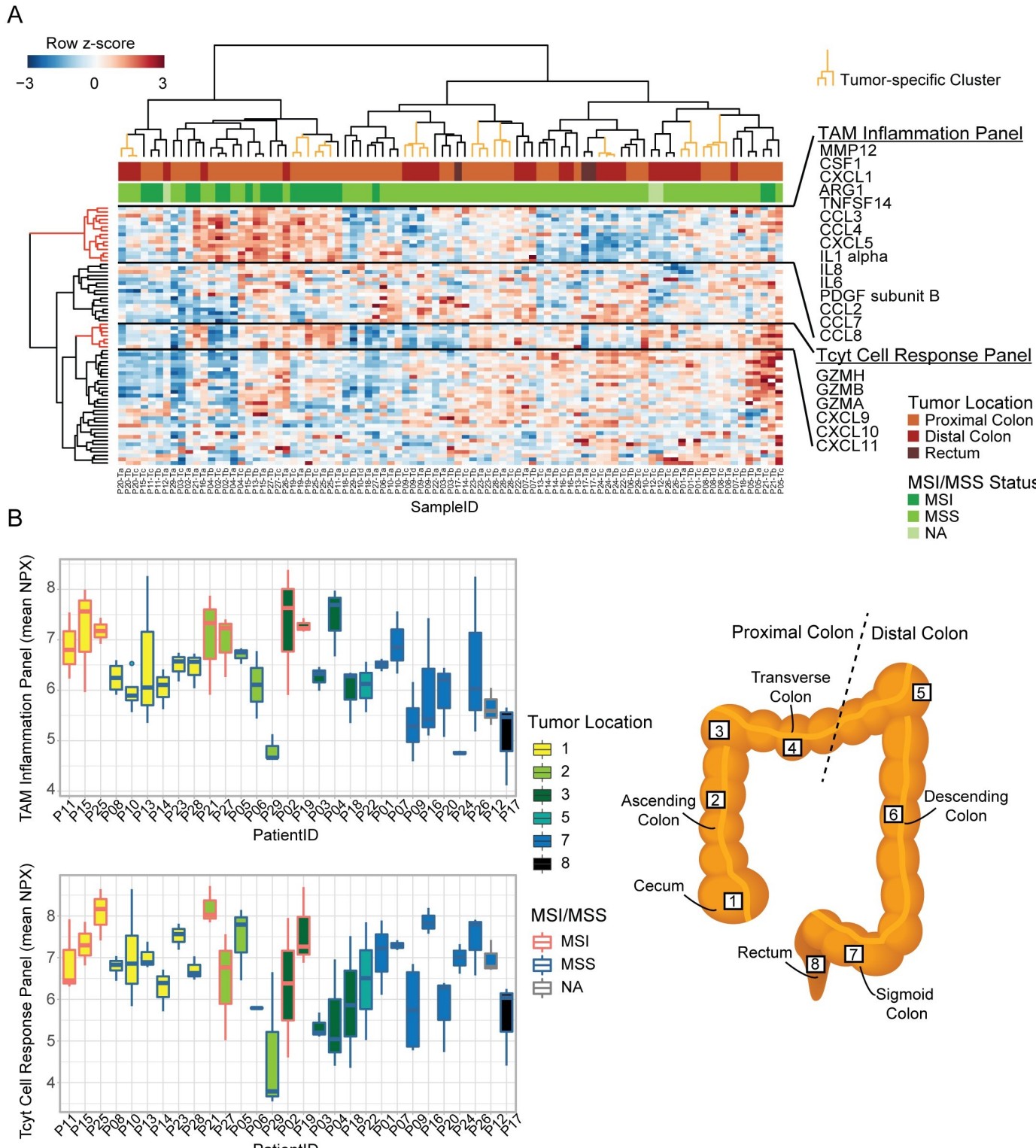

**Fig 4. ITH of immune response on protein level.** (A) Heatmap of protein levels based on the immuno-oncology panel. All columns represent a sample; biopsies in tumor-specific clusters are marked with orange (9/29 tumors). Annotations indicate tumor location and MSI/MSS status. Row-side trees marked with red represent a TAM Inflammation Panel and Tcyt Cell Response Panel. (B) Boxplots showing calculated sample-means for TAM Inflammation panel (top) and the Tcyt cell panel (bottom) for each tumor. Fill colors (1–8) indicate tumor location as illustrated in the schematic figure of the colon and rectum. Red/blue border colors indicate MSI/MSS status. Each bar illustrates results from all biopsies from each tumor. (TAM = tumor associated macrophage, Tcyt = cytotoxic T cell).

## Inflammation at protein level varies between transcriptional subtypes

Multiregional biopsy analysis revealed that some tumors show large variation between biopsies in regards to the TAM inflammation on protein level (Fig 4B). We wanted to see whether this ITH on protein level was related to the transcriptional subtypes (Fig 5A). For the CMS classifier, the CMS1 classified biopsies showed high levels of inflammation, even within the highly heterogeneous tumor P13 (Fig 5A–left). However, for the remaining subtypes (CMS2 and CMS3), the inflammation levels were more similar. This was also the case for the TT classifier, where the biopsies with the highest levels of inflammation were classified as immune-related

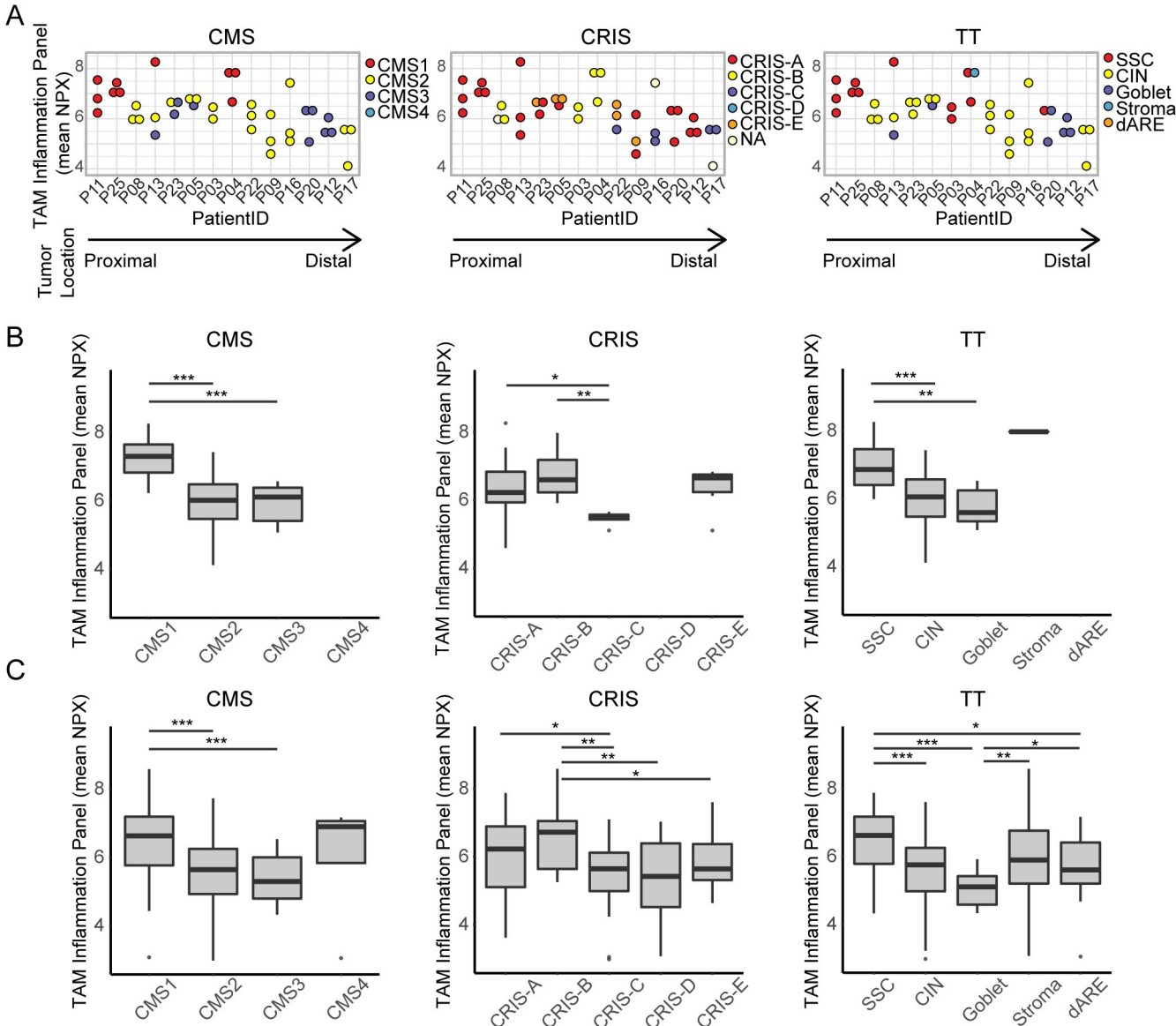

**Fig 5. TAM Inflammation on protein level varies between subtypes.** (A) Dotplots showing TAM inflammation panel (mean NPX) protein level for multiregional biopsies from each tumors (n = 14) with all three classifiers (CMS, CRIS, and TT). Colors indicate transcriptional subtype. Proximal to distal tumor location is indicated by an arrow. (B) Boxplots for each classifier (CMS, CRIS, TT), showing protein inflammation panel mean for all samples (n = 41) grouped based on transcriptional subtype. (C) Boxplots for each classifier, showing protein inflammation panel mean for samples from a validation cohort (n = 162), grouped based on transcriptional subtype. (* = p<0.05; ** = p<0.01; *** = p<0.001, Wilcoxon rank sum test).

subtypes (SSC and Stroma) (Fig 5A–right). In contrast, the immune-related CRIS subtypes (CRIS-A and CRIS-B) showed varying inflammation levels (Fig 5A–middle).

Next, we wanted to see, whether TAM inflammation on protein level was related to the transcriptional subtypes if we grouped all biopsies based on their subtype. The protein level was significantly higher in the subtypes related to immune signaling, including CMS1, CRIS-A, CRIS-B, SSC, and Stroma (p<0.05, Wilcoxon rank rum test) (Fig 5B). This was furthermore verified in an independent validation cohort with single biopsies from 162 stage II and III CRC tumors (Fig 5C). Supporting the notion that the transcriptional based subtypes are linked to the inflammation on protein level.

## Discussion

Most previous studies about ITH of CRC have been focusing on genetic ITH of cancer cells [18–20]. Here we focused on the transcriptional ITH to explore the degree of ITH and the origin thereof, and whether tumor location influence the ITH. Here the presence or absence of ITH was assessed by subtype classification of multiregional tumor biopsies using the well-established CRC subtype classifiers CMS, TT, and CRIS. They reflect different kinds of ITH; either ITH arising due to differences in cell type distributions (CMS and TT) or ITH arising due to transcriptional differences between the cancer cells in the multiregional biopsies (CRIS). Interestingly, we observed that the degree of ITH of subtyping varied depending on tumor location (Fig 1). This was especially pronounced for the CMS classifier, which called half of the proximal tumors with discordant subtypes, while all multiregional biopsies of the distal tumors were called concordantly. The same tendency was observed for the TT classifier, while the CRIS classifier showed the opposite pattern. The CRIS classifications were often discordant within the distal tumors, which match the increased level of high ITH-score genes related to CNA in the distal tumors (Fig 2). We have previously shown that CRIS subtypes are sensitive to CNAs, and that ITH on CNA level influenced the CRIS type called [18], which might be part of the explanation.

We found that clustering of the total transcriptomic profile of protein coding genes lead to tumor-specific clustering for 11 out of 14 tumors (Fig 2A). This indicates that the variation between tumors often exceeds the transcriptomic ITH. A recent study, showed similar results for non-small cell lung cancer, where the majority of samples clustered tumor-specifically [41]. However, for a few tumors we observed that the ITH on transcriptional level was so large, that the samples resembled other tumors more than they resembled each other. To look more into the source of this ITH, we calculated tumor specific ITH-scores for each gene. A large proportion of genes of stromal origin had high ITH-scores, indicating that the stromal TME contributes to a part of the ITH in all tumors (Fig 2B). This heterogeneity of the stromal compartment existed even though we sampled multiregional biopsies from histologically similar tumor areas. Even larger variations have been reported, when comparing different areas such as the central tumor and the invasive front of the tumor [6]. For the most heterogeneous tumors, we created tumor-specific network maps based on ssGSEA terms (Fig 3). Here we saw that gene sets related to metabolism, cell cycle, Wnt signaling, and immune responses varied in activity between the multiregional biopsies. Importantly, these variations matched the subtypes called for each biopsy, which was furthermore related to inflammation on protein level (Fig 5). This indicates that the heterogeneously called subtypes within tumors are actually due to molecular differences between the sampled sites. This supports the view that ITH of subtyping might not be a flaw in the classifier, but rather the reality of the given tumors.

A higher number of discordant called tumors (by CMS and TT) in the proximal colon indicate that the TME may vary more in the proximal tumors compared to the distal tumors. To

look further into this, we analyzed the immune signaling on protein level, and found varying ITH within tumors, and different levels of immune response related to tumor location (Fig 4). Some tumors had very uniform immune signaling in all biopsies, while others showed large variation between sampling sites. These findings suggests that local immune environments exists within the tumors, and that a single biopsy might not suffice, when determining the prognostic impact of the immune response. A recent study by Cremonesi and colleagues highlighted the importance of the microbiota for stimulating chemokine production from cancer cells [42]. They showed that the microbiota influenced T cell trafficking to the tumor, and thereby influenced the prognosis of the patients. One may speculate if the microbiota also exhibits ITH in CRC complicating things even further. In relation to T cell trafficking, we saw that tumors from the cecum exhibited high levels of Tcyt cell response on protein level. This could be due to the immune environment normally present in the cecum [40], which again might be linked to the microbiota. Unfortunately, it is beyond the scope of this study to explore this association further. However, we generally see higher inflammation in the proximal colon MSS tumors, compared to the distal colon MSS tumors (Fig 4).

The MSI tumors analyzed had the highest levels of TAM related inflammation, independent of tumor location even within the proximal colon. While for the MSS/CIN tumors a higher inflammation level was often observed in the proximal colon compared to the distal colon and rectum (Fig 4). In depth characterization of CRC tumors have previously shown that MSS/CIN tumors often have similar CNA profiles independent of tumor location in the proximal or distal colon [16]. Taken together, this may indicate that the location alone influences the inflammatory response in these MSS tumors. Interestingly, several tumors were highly heterogeneous in this regard. Further studies, looking more into the link between ITH, immune response, and tumor location would be interesting. It remains to be seen whether the dichotomy of distinguishing between the proximal and distal colon will eradicate some heterogeneity, or if a more fluent gradient is present along the colon, as suggested by Yamauchi and colleagues [43].

## Conclusion

Our results indicate that ITH of CRC is influenced by the tumor location within the colorectum. By classifying the biopsies into subtypes, we found that the microenvironment more often lead to transcriptional ITH in the proximal colon compared to the distal colon. Importantly, the subtyping heterogeneity between biopsies seem to be due to actual ITH within tumors, since the discordant subtypes matched the biological processes within the biopsies, and the inflammation on protein level. In contrast, the distal tumors were primary classified as discordant due to cancer cell related heterogeneity. Hence, tumor subtyping based on a single biopsy may turn out to be problematic, as tumor location influence the transcriptional ITH in CRC, a topic that may be explored further in future studies. Overall, our results presented here suggest that tumor location, inter-tumor heterogeneity, and ITH should preferably be considered together in future attempts to establish clinically relevant biomarkers for CRC.

## Supporting information

**S1 Fig. Inter-sample correlations in RNA expression and protein expression profiles.** (A) Correlation matrix showing the pair-wise, inter-sample correlations (Pearson's r) in RNA expression profiles (12,593 RNAs with average expression >1 (log2(CPM))). (B) Plot showing the intra-tumor correlation in RNA expression between biopsies (Pearson's r; Y-axis) according to the number of classifiers (CMS, CRIS, TT) that exhibit discordant subtype calls for a tumor sample (X-axis; no discordant calls is '0', whereas values 1, 2 and 3 indicate the number

of classifiers that exhibit discordant subtype calls for a tumor). Biopsies that have discordant calls exhibits lower correlation in RNA expression to the other biopsies from the same tumor than biopsies with no discordant calls. Red bar indicate average values for each category. The p-value indicates that biopsies with no discordant calls for any classifier (X-axis = 0) exhibit significantly higher intra-tumor correlation in RNA expression than biopsies from tumors with discordant classifier calls (X-axis = 1–3) as evaluated by a Wilcoxon rank sum test (WRS). (C) Correlation matrix showing the pair-wise, inter-sample correlations (Pearson's r) in protein expression profiles (68 proteins).
(TIF)

**S2 Fig. Tumor specific network maps.** (A) Subtyping results for each biopsy from tumors P04 and P17. (B-C) Tumor-specific network maps for two tumors illustrating the 5000 ssGSEA terms with the highest ITH for tumor P04 in (B) and tumor P17 in (D). Yellow dots/font indicates mechanisms that are upregulated in the sample compared to the other samples from the same tumor, while blue indicates downregulated mechanisms.
(TIF)

**S1 Table. Sample overview.** Sample information about multiregional biopsies, including PatientID, SampleID, Tumor_location, MSI_MSS, Age, Gender, Subtyping results, and RNA sequencing read depth.
(XLSX)

# Acknowledgments

We thank Pamela Celis, Jesper B. Kristensen, Lisbet Kjeldsen, and Susie L. Larsen for excellent technical assistance. Furthermore, we thank the staff at the NGS Core Center at Aarhus University Hospital, Denmark and the Analysis Service at Olink Proteomics, Uppsala, Sweden. The Danish Cancer Biobank is acknowledged for providing biological material.

# Author Contributions

**Conceptualization:** Sigrid Salling Árnadóttir, Jesper Bertram Bramsen, Claus Lindbjerg Andersen.

**Data curation:** Sigrid Salling Árnadóttir, Trine Block Mattesen, Mogens Rørbæk Madsen, Anders Husted Madsen, Jesper Bertram Bramsen, Claus Lindbjerg Andersen.

**Formal analysis:** Sigrid Salling Árnadóttir, Søren Vang, Nicolai Juul Birkbak, Jesper Bertram Bramsen.

**Funding acquisition:** Sigrid Salling Árnadóttir, Jesper Bertram Bramsen, Claus Lindbjerg Andersen.

**Methodology:** Sigrid Salling Árnadóttir.

**Project administration:** Sigrid Salling Árnadóttir.

**Supervision:** Jesper Bertram Bramsen, Claus Lindbjerg Andersen.

**Visualization:** Sigrid Salling Árnadóttir, Jesper Bertram Bramsen.

**Writing – original draft:** Sigrid Salling Árnadóttir, Jesper Bertram Bramsen, Claus Lindbjerg Andersen.

**Writing – review & editing:** Sigrid Salling Árnadóttir, Trine Block Mattesen, Søren Vang, Mogens Rørbæk Madsen, Anders Husted Madsen, Nicolai Juul Birkbak, Jesper Bertram Bramsen, Claus Lindbjerg Andersen.

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
