## [Decision Letter · Decision Letter 0]

10 Aug 2020

PONE-D-20-20744

Transcriptomic and proteomic intra-tumor heterogeneity of colorectal cancer varies depending on tumor location within the colorectum

PLOS ONE

Dear Dr. Andersen,

Thank you for submitting your manuscript to PLOS ONE. After careful consideration, we feel that it has merit but does not fully meet PLOS ONE’s publication criteria as it currently stands. Therefore, we invite you to submit a revised version of the manuscript that addresses the points raised during the review process.

1.  Statistical significance/meaning is missing from multiple of the data illustrated in the figures.  For the data in figure 2 perform a bootstrap analysis to determine if the heatmap/dendrogram clustering are significant.  For the data in Figure 4 perform a correlation coefficient analysis.

2.  Per both reviewers there are multiple places in which more details are needed.  For example, the total reads for RNA-seq are needed.  Include the other details the reviewers have requested. 

3.  Clarify reviewer 1's question on whether ITH was due to the different transcripts the classifiers were using or were inherent in the data, perhaps this could be made clearer?

4.  Correct the typographical errors and fix the number inconsistencies identified by the reviewers.

5.  Have a collaborator or a scientific writing service read the manuscript and provide suggestions on how to improve readability. 

6. Confirm that the RNA-seq data have been deposited into a repository for access and include the access information.

We look forward to receiving your revised manuscript.

Kind regards,

Amanda Ewart Toland, Ph.D.

Academic Editor

PLOS ONE

Journal Requirements:

2.

We note that you have indicated that data from this study are available upon request. PLOS only allows data to be available upon request if there are legal or ethical restrictions on sharing data publicly. For more information on unacceptable data access restrictions, please see http://journals.plos.org/plosone/s/data-availability#loc-unacceptable-data-access-restrictions.

Reviewers' comments:

Reviewer's Responses to Questions

**Comments to the Author**

1. Is the manuscript technically sound, and do the data support the conclusions?

Reviewer #1: Yes

Reviewer #2: Partly

2. Has the statistical analysis been performed appropriately and rigorously? 

Reviewer #1: Yes

Reviewer #2: No

3. Have the authors made all data underlying the findings in their manuscript fully available?

Reviewer #1: Yes

Reviewer #2: Yes

4. Is the manuscript presented in an intelligible fashion and written in standard English?

Reviewer #1: Yes

Reviewer #2: Yes

5. Review Comments to the Author

Reviewer #1: Árnadóttir et al, have RNA sequenced 41 biopsies from 14 stage II or III CRC tumours. They also carried out multiplex immune protein analysis on 89 biopsies from 29 tumours. The amount of ITH was explored by analysing the RNA sequencing data from multi-region samples. By combining three different RNA sequencing based classifiers they hoped to establish an understanding on the amount of ITH present. Specifically, they combined CRC Intrinsic Subtypes (CRIS), The Consensus Molecular Subtypes (CMS) and TUMOR type (TT). This was an interesting paper to read. Below are some minor points:

The paper has been written well and is clear, with only two minor points. There was one typo present on line 51 “outcome [2] In” is missing a full stop. A grammatical mistake was present on line 393 "It might be speculated, whether the microbiota exhibit ITH in CRC complicating things even further."

Regarding the work, it was a little difficult to discern whether or not the ITH was due to the different transcripts the classifiers were using or were inherent in the data, perhaps this could be made clearer?

Regarding the statistical analysis, one comment regarding the statistical analysis, on the heatmap and dendrogram on Figure 2a - it might be useful to also bootstrap to determine whether or not the clusters are significant.

Reviewer #2: The manuscript by Arnadottir et al describes an interesting multiregional sampling of multiple tumors (14 tumors with 41 regions for RNA-seq; 29 tumors with 89 regions for multiplex immunoassays). Similar DNA sequencing types of studies have revealed widespread genetic and epigenetic ITH. The authors find that sometimes the regions are concordant for the measurements within a tumor, and sometimes not. They show that tumor location within the colorectum influence the type of expression ITH. Overall, these findings are of interest. The manuscript is complex and difficult to follow in many places. This reviewer found it difficult to get a “big picture” impression of the data. However, the overall data are important because they show how one biopsy may not be adequate for reproducible classification using the illustrated schemes. Several comments:

1) Fig 1 presents the overall data well. However, the Results for the RNA-seq data (page 8) are based on classification schemes, with 50% of the tumors having at least one biopsy with a different classification. It would be useful to:

a) provide the RNA-seq depth (reads per sample)

b) provide some statistical summary of the raw data (ie the correlation of CPM values between samples from the same tumor)

c) Provide the 50% misclassification rate information in the Abstract

2) Fig 2 shows that using RNA expression levels, the different biopsies from the same patients tend to cluster with individual patients. This Figure is hard to decipher. It might be useful to put arrows to indicate samples that do not cluster by patient (ie such as P05-Tc). The manuscript also states that samples “cluster” by subtype, but this claim is not all that obvious in Fig 2A because some subtypes (say for example CRIS) seem more randomly distributed along the horizontal axis. This statement should be clarified.

3) Fig 3 looks at the most discordantly expressed genes (STD>.5) and does a supervised gene enrichment analysis. Such types of analysis almost always give an output. The authors describe their analysis but provide little insights on how this type of analysis can explain expression ITH.

4) Fig 4 show abundant protein level ITH with only 9 of 30 tumors (why is it 29 in the Abstract?) having its multiple regions clustering together. A simple, more standard statistical description such as a correlation coefficient (within and between tumors) could provide more meaning to this data. This poor concordance information between biopsies from the same tumor should be mentioned in the Abstract.

6. PLOS authors have the option to publish the peer review history of their article (what does this mean?). If published, this will include your full peer review and any attached files.

Reviewer #1: No

Reviewer #2: No

---

## [Author Response · Author response to Decision Letter 0]

26 Sep 2020

PONE-D-20-20744

Transcriptomic and proteomic intra-tumor heterogeneity of colorectal cancer varies depending on tumor location within the colorectum

PLOS ONE

Dear Dr. Andersen,

Thank you for submitting your manuscript to PLOS ONE. After careful consideration, we feel that it has merit but does not fully meet PLOS ONE’s publication criteria as it currently stands. Therefore, we invite you to submit a revised version of the manuscript that addresses the points raised during the review process.

Response from Authors: We are indeed very happy that PLOS ONE finds the manuscript to have merit for publication. As described below, we have addressed the comments made by the reviewers and we would like to thank both the editor and reviewers for their constructive remarks.

1. Statistical significance/meaning is missing from multiple of the data illustrated in the figures. For the data in figure 2 perform a bootstrap analysis to determine if the heatmap/dendrogram clustering are significant. For the data in Figure 4 perform a correlation coefficient analysis.

Response from Authors: A bootstrap analysis for the data in Figure 2 and a coefficient analysis for the data in Figure 4 (new S1 Fig) have been added as requested. For further details, please see the responses to the reviewers below.

2. Per both reviewers there are multiple places in which more details are needed. For example, the total reads for RNA-seq are needed. Include the other details the reviewers have requested.

Response from Authors: RNA-seq read depths (total reads) have been added to the excel table ‘S1_Table.xlsx’. Please also see our responses below.

3. Clarify reviewer 1's question on whether ITH was due to the different transcripts the classifiers were using or were inherent in the data, perhaps this could be made clearer?

Response from Authors: This question has been addressed, as stated below in the response to the reviewer’s question (please see below). The following sentences have been added:

Line 197: ‘Since these classifiers use different gene sets and approaches for subtyping, they reflect different kinds of ITH; e.g. ITH arising due to inter-biopsy differences in cell type distribution of both cancer epithelium and tumor stromal cells (CMS and TT) or ITH arising due to transcriptional differences between the cancer cell populations in the biopsies (CRIS).’

Line 380: ‘Here the presence or absence of ITH was assessed by subtype classification of multiregional tumor biopsies using the well-established CRC subtype classifiers CMS, TT and CRIS. They reflect different kinds of ITH; either ITH arising due to differences in cell type distributions (CMS and TT) or ITH arising due to transcriptional differences between the cancer cells in the multiregional biopsies (CRIS).’ 

4. Correct the typographical errors and fix the number inconsistencies identified by the reviewers.

Response from Authors: These errors have been corrected.

5. Have a collaborator or a scientific writing service read the manuscript and provide suggestions on how to improve readability. 

Response from Authors: The manuscript has been reviewed by a collaborator to ease readability. 

6. Confirm that the RNA-seq data have been deposited into a repository for access and include the access information.

Response from Authors: The RNA-seq data is being deposited into the European Genome-phenome Archive (EGA) database (accession number (EGAS00001004668). As the data are personal and potentially sensitive, the National Committee on Health Research Ethics and the Danish Data Protection Agency requires that both the European general data protection regulation (GDPR) and Danish data protection laws are complied with unconditionally, before data access can be granted. This means that request’ties have to accept to enter into a data access agreement with the Danish data controller, which among other things ensure that the data will only be used for statistical and/or research purposes and that confidentiality of the data subjects will preserved at all times. We confirm that the authors had no special access to data, and that qualifying readers can obtain the same data the authors had access to.

Data availability statement:

 ‘RNA sequencing data is deposited at the European Genome-phenome Archive (EGA, [https://www.ebi.ac.uk/ega/] accession number EGAS00001004668, which is hosted by the European Bioinformatics Institute (EBI) and the Centre for Genomic Regulation (CRG). Data sharing is only possible for the sole purpose of carrying out statistical or scientific research of significant importance to society. Request for data access should be directed to the EGAS00001004668 Data Access Committee through the EGA website. 

5. Review Comments to the Author

Reviewer #1: Árnadóttir et al, have RNA sequenced 41 biopsies from 14 stage II or III CRC tumours. They also carried out multiplex immune protein analysis on 89 biopsies from 29 tumours. The amount of ITH was explored by analysing the RNA sequencing data from multi-region samples. By combining three different RNA sequencing based classifiers they hoped to establish an understanding on the amount of ITH present. Specifically, they combined CRC Intrinsic Subtypes (CRIS), The Consensus Molecular Subtypes (CMS) and TUMOR type (TT). This was an interesting paper to read. Below are some minor points:

The paper has been written well and is clear, with only two minor points. There was one typo present on line 51 “outcome [2] In” is missing a full stop. A grammatical mistake was present on line 393 "It might be speculated, whether the microbiota exhibit ITH in CRC complicating things even further."

Response from Authors: 

A full stop has been added in line 51.

The grammatical error has been corrected, the sentence now reads (Line 418): ‘One may speculate if the microbiota also exhibits ITH in CRC complicating things even further.’

Regarding the work, it was a little difficult to discern whether or not the ITH was due to the different transcripts the classifiers were using or were inherent in the data, perhaps this could be made clearer?

Response from Authors: The origin of the ITH has now been explained further to highlight that the difference in ITH observed by the three classifiers as discordant subtype calls, is due to the different transcripts in the classifiers. Overall, the observed ITH between biopsies are reflected by many transcripts: we now present a correlation matrix in S1A Fig and a summary figure in S1B Fig, which illustrate that biopsies from patients with discordant subtype calls exhibit a poorer intra-patient correlation in RNA expression (>12.000 RNAs) than patients with concordant subtype calls. This indicates that ITH affects many transcripts and goes beyond the transcripts that are included in each of the classifiers. Still, among the many transcripts that exhibit ITH, the different classifiers focus on different transcript subsets. The CRIS classifier was originally developed by selecting only transcript originating from the cancer epithelial cells. Therefore, CRIS classification will primarily capture the ITH caused by different cancer cell clones in the multiregional biopsies. In contrast, the CMS and TT classifiers were developed using all transcripts (both from the epithelial cells and the surrounding tumor stroma cells). Consequently, CMS and TT are additionally capturing ITH that is arising due to differences in the abundance of different non-tumor (e.g. immune) cell types in the biopsy, in addition to differences in the cancer cells. In accordance with these differences in the classifier transcripts we find that the CMS classifier exhibit discordant call exclusively in the proximal colon, which is known to exhibit higher and varying immune cell infiltration (e.g PMCID: PMC6048410), whereas the CRIS classifier exhibit most discordance in the immune cell poorer distal/rectal tumors, where cancer epithelial transcripts are expected to contribute relatively more to ITH. To introduce these considerations in the manuscript the following sentences have been added:

Line 197: ‘Since these classifiers use different gene sets and approaches for subtyping, they reflect different kinds of ITH; e.g. ITH arising due to inter-biopsy differences in cell type distribution of both cancer epithelium and tumor stromal cells (CMS and TT) or ITH arising due to transcriptional differences between the cancer cell populations in the biopsies (CRIS).’

Line 232: ‘Hence, the ITH detected with the classifiers, which use a subset of transcripts, are also present when assessing ITH across all transcripts. In line with this, tumors that were concordantly subtyped had a significantly higher intra-tumor correlation at transcriptional level, compared to tumors with discordantly subtyped biopsies (p = 0.0037, Wilcoxon rank sum test) (S1B Fig).’ 

Line 380: ‘Here the presence or absence of ITH was assessed by subtype classification of multiregional tumor biopsies using the well-established CRC subtype classifiers CMS, TT and CRIS. They reflect different kinds of ITH; either ITH arising due to differences in cell type distributions (CMS and TT) or ITH arising due to transcriptional differences between the cancer cells in the multiregional biopsies (CRIS).’

Regarding the statistical analysis, one comment regarding the statistical analysis, on the heatmap and dendrogram on Figure 2a - it might be useful to also bootstrap to determine whether or not the clusters are significant.

Response from Authors: A bootstrap analysis has been performed for the data in figure 2a and the figure has been updated to with information on the significance of tumor-specific clustering. The following information has been added to the material and methods:

Line 156: ‘Clustering bootstrapping was performed to evaluate the significance of patient-specific clusters using the pvclust R package (1000 repetitions) and significance were estimated by Approximately Unbiased (AU) p-values: Clusters were considered significant for AU values ≥95, which indicates a significance p-value ≤0.05 [30].’ 

And in the figure legend for figure 2a:

Line 249: ‘The clusters indicated with asterisk (*) were statistical significant as evaluated by bootstrapping (Approximately Unbiased (AU) values ≥95).’

Reviewer #2: The manuscript by Arnadottir et al describes an interesting multiregional sampling of multiple tumors (14 tumors with 41 regions for RNA-seq; 29 tumors with 89 regions for multiplex immunoassays). Similar DNA sequencing types of studies have revealed widespread genetic and epigenetic ITH. The authors find that sometimes the regions are concordant for the measurements within a tumor, and sometimes not. They show that tumor location within the colorectum influence the type of expression ITH. Overall, these findings are of interest. The manuscript is complex and difficult to follow in many places. This reviewer found it difficult to get a “big picture” impression of the data. However, the overall data are important because they show how one biopsy may not be adequate for reproducible classification using the illustrated schemes. 

Several comments:

1) Fig 1 presents the overall data well. However, the Results for the RNA-seq data (page 8) are based on classification schemes, with 50% of the tumors having at least one biopsy with a different classification. It would be useful to:

a) provide the RNA-seq depth (reads per sample)

Response from Authors: The RNA-seq read depths (total reads per sample) have been added to the excel file ‘S1_Table.xlsx’.

Moreover the following sentence has been added to the manuscript.

Line 118: ‘A minimum of 34 million read pairs (median 65 million read pairs) were sequenced per sample on an Illumina NextSeq500 using high output flow cells (Illumina).’

b) provide some statistical summary of the raw data (ie the correlation of CPM values between samples from the same tumor)

Response from Authors: We have included correlation matrixes as S1 Fig, which contain the Pearson’s correlation r-values for pairwise inter-sample comparisons of CPM values from all 41 samples with RNA data and protein expression values for 89 samples with protein data.

The following lines have been added to the manuscript:

Line 230: ‘…pair-wise inter-sample correlations in RNA expression profiles are given in S1A Fig.’

Line 322: ‘…pair-wise inter-sample correlations in protein expression profiles are given in S1C Fig).’

c) Provide the 50% misclassification rate information in the Abstract

Response from Authors: This information has been added to the abstract (line 35), which now reads: ‘Subtyping of proximal MSS tumors were discordant for 50% of the tumors, this ITH was related to differences in the microenvironment. Subtyping of distal MSS tumors were less discordant, here the ITH was more cancer-cell related’.

2) Fig 2 shows that using RNA expression levels, the different biopsies from the same patients tend to cluster with individual patients. This Figure is hard to decipher. It might be useful to put arrows to indicate samples that do not cluster by patient (ie such as P05-Tc). The manuscript also states that samples “cluster” by subtype, but this claim is not all that obvious in Fig 2A because some subtypes (say for example CRIS) seem more randomly distributed along the horizontal axis. This statement should be clarified.

Response from Authors: Arrows have been added to Fig 2, indicating the location of the samples from the heterogeneous tumors P05, P13 and P17. The following description has been added to the figure legend (line 250): ‘Arrows below the heatmap indicate discordant clustering biopsies (black: P05, red: P13, blue: P17).’ 

The statement about clustering of subtypes has been clarified (line 236), it now reads: ‘…However, the majority of biopsies cluster in tumor-specific clusters. Furthermore, the biopsies also tend to cluster based on the subtype combination called across the classifiers. This is especially pronounced for the CMS and TT subtypes while the CRIS subtypes are more intermixed. For some heterogeneous tumors (P05 and P13), their biopsies cluster with biopsies from other tumors with the same subtype combination, rather than clustering in tumor-specific clusters.’

3) Fig 3 looks at the most discordantly expressed genes (STD>.5) and does a supervised gene enrichment analysis. Such types of analysis almost always give an output. The authors describe their analysis but provide little insights on how this type of analysis can explain expression ITH.

Response from Authors: The following sentence has been added (line 308): ‘These network maps illustrates the dynamic state of RNA transcripts, as differences in ongoing cellular functions cause ITH on RNA level.’

4) Fig 4 show abundant protein level ITH with only 9 of 30 tumors (why is it 29 in the Abstract?) having its multiple regions clustering together. A simple, more standard statistical description such as a correlation coefficient (within and between tumors) could provide more meaning to this data. This poor concordance information between biopsies from the same tumor should be mentioned in the Abstract.

Response from Authors: ‘9 out of 30’ is an error, it has been corrected to ‘9 out of 29’.

We have included correlation matrixes as S1 Fig, which contain the Pearson’s correlation r-values for pairwise inter-sample comparisons of CPM values from all 41 samples with RNA data and protein expression values for 89 samples with protein data.

The information about the poor concordance on protein level between biopsies from the same tumor has been added to the abstract.

The following lines have been added to the manuscript:

Line 230: ‘…pair-wise inter-sample correlations in RNA expression profiles are given in S1A Fig.’

Line 322: ‘…pair-wise inter-sample correlations in protein expression profiles are given in S1C Fig).’

Line 40 (abstract): ‘Unsupervised hierarchical clustering of the protein data identified large ITH at protein level; as the multiregional biopsies clustered together for only 9 out of 29 tumors.’

---

## [Decision Letter · Decision Letter 1]

9 Oct 2020

Transcriptomic and proteomic intra-tumor heterogeneity of colorectal cancer varies depending on tumor location within the colorectum

PONE-D-20-20744R1

Dear Dr. Andersen,

We’re pleased to inform you that your manuscript has been judged scientifically suitable for publication and will be formally accepted for publication once it meets all outstanding technical requirements.

Kind regards,

Amanda Ewart Toland, Ph.D.

Academic Editor

PLOS ONE

Additional Editor Comments (optional):

Reviewers' comments:

Reviewer's Responses to Questions

**Comments to the Author**

1. If the authors have adequately addressed your comments raised in a previous round of review and you feel that this manuscript is now acceptable for publication, you may indicate that here to bypass the “Comments to the Author” section, enter your conflict of interest statement in the “Confidential to Editor” section, and submit your "Accept" recommendation.

Reviewer #1: All comments have been addressed

Reviewer #2: All comments have been addressed

2. Is the manuscript technically sound, and do the data support the conclusions?

Reviewer #1: Yes

Reviewer #2: Yes

3. Has the statistical analysis been performed appropriately and rigorously? 

Reviewer #1: Yes

Reviewer #2: Yes

4. Have the authors made all data underlying the findings in their manuscript fully available?

Reviewer #1: Yes

Reviewer #2: Yes

5. Is the manuscript presented in an intelligible fashion and written in standard English?

Reviewer #1: Yes

Reviewer #2: Yes

6. Review Comments to the Author

Reviewer #1: (No Response)

Reviewer #2: The authors have improved the manuscript. Describing "heterogeneity" is complex and the authors have done a good job in describing ITH at the protein and transcript level in CRCs

7. PLOS authors have the option to publish the peer review history of their article (what does this mean?). If published, this will include your full peer review and any attached files.

Reviewer #1: No

Reviewer #2: No

---

## [Editor Report · Acceptance letter]

9 Dec 2020

PONE-D-20-20744R1 

Transcriptomic and proteomic intra-tumor heterogeneity of colorectal cancer varies depending on tumor location within the colorectum 

Dear Dr. Andersen:

I'm pleased to inform you that your manuscript has been deemed suitable for publication in PLOS ONE. Congratulations! Your manuscript is now with our production department. 

Kind regards, 

on behalf of

Dr. Amanda Ewart Toland 

Academic Editor

PLOS ONE